# Transcriptome Profiling of Spike Development Reveals Key Genes and Pathways Associated with Early Heading in Wheat–*Psathyrstachys huashanica* 7Ns Chromosome Addition Line

**DOI:** 10.3390/plants14132077

**Published:** 2025-07-07

**Authors:** Binwen Tan, Yangqiu Xie, Hang Peng, Miaomiao Wang, Wei Zhu, Lili Xu, Yiran Cheng, Yi Wang, Jian Zeng, Xing Fan, Lina Sha, Haiqin Zhang, Peng Qin, Yonghong Zhou, Dandan Wu, Yinghui Li, Houyang Kang

**Affiliations:** 1State Key Laboratory of Crop Gene Exploration and Utilization in Southwest China, Sichuan Agricultural University, Chengdu 611130, China; yangli950917@163.com (B.T.); xyq0097email@163.com (Y.X.); 13808181840@163.com (H.P.); wangmiaomiao0220@163.com (M.W.); zhuwei202209@163.com (W.Z.); chengyiran@sicau.edu.cn (Y.C.); wangyi@sicau.edu.cn (Y.W.); fanxing9988@163.com (X.F.); zhouyh@sicau.edu.cn (Y.Z.); wudandan@sicau.edu.cn (D.W.); 2Triticeae Research Institute, Sichuan Agricultural University, Chengdu 611130, China; xulili_0627@126.com; 3College of Resources, Sichuan Agricultural University, Chengdu 611130, China; zengjian@sicau.edu.cn; 4College of Grassland Science and Technology, Sichuan Agricultural University, Chengdu 611130, China; rice_shazhi@163.com (L.S.); haiqinzhang@163.com (H.Z.); 5College of Agronomy and Biotechnology, Yunnan Agricultural University, Kunming 650201, China; wheat-quinoa@ynau.edu.cn

**Keywords:** *Psathyrostachys huashanica*, spike development, heading time, early maturation, transcription factors

## Abstract

Developing early-heading wheat cultivars is an important breeding strategy to utilize light and heat resources, facilitate multiple-cropping systems, and enhance annual grain yield. *Psathyrostachys huashanica* Keng (2*n* = 2*x* = 14, NsNs) possesses numerous agronomically beneficial traits for wheat improvement, such as early maturity and resistance to biotic and abiotic stresses. In this study, we found that a cytogenetically stable wheat–*P. huashanica* 7Ns disomic addition line showed (9–11 days) earlier heading and (8–10 days) earlier maturation than its wheat parents. Morphological observations of spike differentiation revealed that the 7Ns disomic addition line developed distinctly faster than its wheat parents from the double ridge stage. To explore the potential molecular mechanisms underlying the early heading, we performed transcriptome analysis at four different developmental stages of the 7Ns disomic addition line and its wheat parents. A total of 10,043 differentially expressed genes (DEGs) were identified during spike development. Gene Ontology (GO) enrichment analysis showed that these DEGs were linked to the carbohydrate metabolic process, photosynthesis, response to abscisic acid, and the ethylene-activated signaling pathway. Kyoto Encyclopedia of Genes and Genomes (KEGG) enrichment analysis showed that these DEGs were involved in plant hormone signal transduction (*ARF*, *AUX/IAA*, *SAUR*, *DELLA*, *BRI1*, and *ETR*), starch and sucrose metabolism (*SUS1* and *TPP*), photosynthetic antenna proteins (*Lhc*), and circadian rhythm (*PRR37*, *FT*, *Hd3a*, *COL*, and *CDF*) pathways. In addition, several DEGs annotated as transcription factors (TFs), such as bHLH, bZIP, MADS-box, MYB, NAC, SBP, WRKY, and NF-Y, may be related to flowering time. Our findings reveal spike development-specific gene expression and critical regulatory pathways associated with early heading in the wheat–*P. huashanica* 7Ns addition line, and provide a new genetic resource for further dissection of the molecular mechanisms underlying the heading date in wheat.

## 1. Introduction

Wheat (*Triticum aestivum* L., 2*n* = 6*x* = 42, AABBDD) is a staple food crop widely cultivated worldwide and provides approximately 20% of the calories consumed by human beings and livestock [1]. Wheat heading time, representing the initiation of flowering time, is an important agronomic trait related to ecological adaptability, maturity, yield, and stress resistance. Early maturity is essential to reduce potential yield losses caused by several negative factors, such as frost, heat, disease, pre-harvest sprouting, and terminal drought stress [2,3]. More importantly, it is beneficial for promoting wheat–rice or wheat–maize rotations in Southwest China. Therefore, the development of early-heading wheat lines and studies on their related candidate genes or pathways are crucial to maximize the whole-year crop yield potential.

Heading time in wheat, a complex polygenic trait, is mainly influenced by vernalization (*VRN*), photoperiod (*PPD*), and earliness per se (*Eps*) [4]. Vernalization is primarily controlled by *VRN1*, *VRN2*, and *VRN3* [5]. *VRN1*, an *APETALA1* (*AP1*)-like MADS-box transcription factor, is the central regulator that can interact with *TaVRT2*, an SVP-like gene, to regulate vernalization-induced flowering [6,7]. *VRN2* acts as a flowering repressor and is downregulated by vernalization and short-day (SD) treatment [8,9]. *VRN3*, a homologous gene of the *Arabidopsis FLOWERING LOCUS T* (*FT*), moves from the leaf to the apical meristem to induce flowering [5]. *PPD1*, a pseudo-response regulator (PRR) gene that controls photoperiod-dependent floral induction, affects inflorescence architecture and paired spikelet formation by modulating the *FT* expression [10,11]. When the vernalization and photoperiod requirements are fully satisfied, flowering time is mainly determined by *Eps* genes, which are independent of environmental cues [12]. To date, only a few of the underlying *Eps* genes have been identified [4,13], but their molecular mechanisms regulating heading time are poorly understood. Additionally, several phytohormones, such as gibberellin (GA), abscisic acid (ABA), brassinosteroid (BR), auxin (IAA), methyl jasmonate (MeJA), and ethylene, also play important roles in fine-tuning the timing of flowering [14,15,16]. Among them, GA plays a major role in affecting flowering time [17]. DELLA proteins, key components in the GA signaling pathway, physically interact with the flowering activator *CONSTANS* (*CO*) to regulate flowering under long day (LD) conditions in *Arabidopsis* [18]. GA promotes the transcription of the MADS-box gene *SUPPRESSOR OF OVEREXPRESSION OF CO1* (*SOC1*) to accelerate flowering in *Arabidopsis* [19]. Both *VRN1* and GA are required in the wheat shoot apical meristem for the acceleration of spike development under SD conditions [20]. Factors contributing to early wheat maturation include the following: (1) accelerated floral transition, characterized by premature conversion of the shoot apical meristem into the spike meristem; (2) expedited spike development with a rapid progression from primordium formation to mature spike; (3) shortened pre-heading phase between spike formation and ear emergence; (4) reduced heading–anthesis interval; and (5) abbreviated grain-filling period from anthesis to physiological maturity. Crucially, accelerated spike development and early phase transitions represent the core physiological determinants of this process [21,22].

Spike development is pivotal for floral organ formation and flower induction in cereals. In recent years, RNA-seq has been extensively employed to investigate the molecular mechanisms regulating flowering time driven by spike development. For instance, Digel et al. [23] performed a transcriptome analysis using developing leaf and shoot apices to reveal the distinct genetic and environmental control of floral transition and inflorescence development in barley. Through transcriptome analysis of early spike development in wheat, Li et al. [24] identified 375 transcription factor genes that are involved in flowering time regulation, meristem initiation or transition, and floral organ development. Liu et al. [25] profiled transcriptomes at three developmental stages of the barley main shoot apex to uncover phase-specific gene expression related to barley inflorescence and identify novel candidate genes that regulate meristem activities and flower development. Transcriptome analysis during the double ridge and androgynous primordium differentiation stages of the wheat leaf and apical meristem revealed many DEGs associated with wheat heading date and identified a potential candidate gene influencing flowering time [26]. Additionally, VanGessel et al. [27] utilized a wheat spike transcriptome dataset to reveal dynamic expression profiles linked to the progression from vegetative meristem formation to terminal spikelet establishment, highlighting potential roles for *TtCLE13*, *TtWOX2*, and *TtWOX7* in wheat meristem development. Benaouda et al. [28] discovered that the wheat orthologous transcription factor *AS1* could induce flowering time in response to GA biosynthesis via transcriptome analysis of the wheat shoot apical meristem and leaf tissue. Gauley et al. [29] performed a transcriptome analysis of wheat developing inflorescence and identified bZIP and ALOG transcription factors, namely PDB1 and ALOG1, which influence flowering time and spikelet architecture.

As an important tertiary gene pool of wheat genetic improvement, *Psathyrostachys huashanica* Keng ex P. C. Kuo (2*n* = 2*x* = 14, NsNs) harbors numerous agronomically beneficial traits, such as early maturity and resistance to biotic and abiotic stresses [30,31]. At present, a large number of genes of interest from *P. huashanica* have been successfully introgressed into common wheat through the generation of a series of wheat–*P. huashanica* derived lines [32,33,34]. Previously, we developed and characterized a wheat–*P. huashanica* 7Ns disomic chromosome addition line 18-1-5 with powdery mildew resistance [35]. In the current study, we investigated the heading and maturity times of 18-1-5 and its wheat parents, Chinese Spring (CS) and CS*ph2b*, under field conditions. Anatomical observation of young spikes was performed to further characterize their phenotypic differences. Moreover, comparative transcriptome analysis between 18-1-5 and its wheat parents was conducted across four different spike development stages to explore the molecular mechanisms underlying accelerated spike development in 18-1-5. Our data provide new insights into the genetic regulatory mechanism of wheat heading time and offer genetic resources for early-maturing wheat breeding.

## 2. Results

### 2.1. Cytological Identification of Wheat–P. huashanica 7Ns Addition Line

To assess the cytogenetic stability of the wheat–*P. huashanica* 7Ns disomic addition line 18-1-5, genomic in situ hybridization (GISH) and fluorescence in situ hybridization (FISH) techniques were employed to characterize 80 randomly selected individual plants from the selfed progeny of 18-1-5. GISH analysis revealed that all examined plants carried 42 chromosomes with blue 4,6-diamidino-2-phenylindole (DAPI) signals and two chromosomes with strong red fluorescent signals that were Ns chromosomes of *P. huashanica* (Figure 1a). FISH analysis using the probes Oligo-pSc119.2 and Oligo-pTa535 demonstrated that all plants contained 21 pairs of intact wheat chromosomes in accordance with the standard FISH karyotype of CS [36] (Figure 1b). Further FISH analysis with the probes Oligo-pSc200, Oligo-44, and Oligo-pTa71A-2 indicated that the additional alien chromosome in all plants was identified as belonging to *P. huashanica* chromosome 7Ns based on the previously published FISH karyotype of *P. huashanica* [37] (Figure 1c). These results suggested that 18-1-5 was a cytogenetically stable wheat–*P. huashanica* 7Ns disomic addition line, which could be utilized for subsequent research.

### 2.2. Investigation of Heading and Maturity Times for Wheat–P. huashanica 7Ns Addition Line

We investigated the heading and maturity times of wheat–*P. huashanica* 7Ns disomic addition line 18-1-5 and its wheat parents CS and CS*ph2b* in the field for three consecutive years. The statistical results revealed that the average heading time of 18-1-5 was significantly shorter than that of wheat parents CS and CS*ph2b*, with an advance of 10.7 and 9.3 days, respectively (Figure 2a,b; Appendix A). Moreover, the average maturity time of 18-1-5 was also significantly earlier than that of CS and CS*ph2b*, with an advance of 10.0 and 8.3 days, respectively (Figure 2c,d; Appendix A).

### 2.3. Observation of Spike Development in the Wheat–P. huashanica 7Ns Addition Line

To further reveal the reason behind the different heading and maturity time between 18-1-5 and its wheat parents, we used a stereomicroscope to observe spike differentiation from the three-leaf stage to the heading stage during the 2022–2023 growing season, and recorded sixteen different developmental time points (Figure 3; Appendix A). The results showed that no significant differences appeared among the materials at the apex elongation stage and early single-ridge stage at 24 and 34 days after sowing, respectively (Figure 3a,b). However, a clear developmental difference emerged at 43 days after sowing, where CS and CS*ph2b* were at the middle single-ridge stage, whereas 18-1-5 had entered the early double-ridge stage (Figure 3c). Thereafter, the spike development rate of 18-1-5 was faster than that of CS and CS*ph2b* from the later double-ridge stage to the tetrad stage (Figure 3d–o). Finally, at 143 days after sowing, CS and CS*ph2b* were at the tetrad stage, while 18-1-5 developed larger spikes and started heading (Figure 3p). Thus, these findings suggested that the spike development process of wheat parents CS and CS*ph2b* was basically consistent, while 18-1-5 reached spike development earlier than its wheat parents during the double-ridge stage to the tetrad stage.

### 2.4. Quality Analysis and Sequence Assembly of RNA-Seq Data

To identify the underlying genes responsible for the early heading time of 18-1-5, the developing young spikes of three materials, including 18-1-5 and its wheat parents CS and CS*ph2b*, were collected at four different spike developmental stages, which were referred to as the double-ridge stage (S1), the glume primordia differentiation stage (S2), the floret primordia differentiation stage (S3), and the stamen and pistil differentiation stage (S4), respectively. A total of 36 RNA samples from four developmental stages of three materials with three biological replicates were subjected to RNA sequencing, and approximately 2.38 billion raw reads were generated. After filtering, approximately 2.36 billion (98.82%) high-quality clean reads were retained. Each library contained 59.23–79.18 million reads. The Q30 value was greater than 97%, and the guanine and cytosine (GC) content distribution was 49–52%. Following assembly, approximately 2.23 billion clean reads were mapped to IWGSC Refseq v1.1, with an average rate of 94.58%, of which 2.10 billion reads could be aligned to only one location on the reference genome, ranging from 84.72 to 91.27% in different samples (Appendix A). The transcriptome data met the quality requirements for subsequent analysis to a high extent.

### 2.5. DEGs Obtained During Spike Development

To obtain differentially expressed genes (DEGs) between 18-1-5 and its wheat parents, CS and CS*ph2b* were pooled together as the parental control since they were similar in spike development. DEGs between 18-1-5 (represented by a capital letter “C”) and its wheat parents (represented by a capital letter “D”) across the four different developmental stages were analyzed using DESeq2 v1.26.0 software. In total, we identified 7189 unique DEGs (Appendix A). In the C-S1 vs. D-S1, C-S2 vs. D-S2, C-S3 vs. D-S3, and S4 vs. D-S4 comparison groups, there were 3490, 1744, 2137, and 2672 DEGs, respectively (Figure 4a). Among them, 2277, 1086, 1005, and 1521 DEGs were unique in the C-S1 vs. D-S1, C-S2 vs. D-S2, C-S3 vs. D-S3, and C-S4 vs. D-S4 comparison groups, respectively, and 257 DEGs, including 59 upregulated and 198 downregulated, were shared in four different comparison groups (Figure 4b,c), suggesting that these genes may be involved in maintaining young spike development.

### 2.6. GO Enrichment Analysis of DEGs

To explore the regulatory pathways of these DEGs, Gene Ontology (GO) enrichment analysis was performed. The results showed that all DEGs were classified into 356 unique GO terms with three categories: biological process (BP), cellular component (CC), and molecular function (MF) (Appendix A). The top 30 GO terms with the most significant enrichment were selected for further analysis (Figure 5). In the C-S1 vs. D-S1 group, the main BPs were protein phosphorylation, transmembrane transport, carbohydrate metabolic process, and photosynthesis; the main CCs were nucleus, chloroplast, and photosystem I/II; and the main MFs were protein binding, DNA binding, and ATP binding (Figure 5a). In the C-S2 vs. D-S2 group, the main BPs were protein phosphorylation, regulation of transcription, transmembrane transport, and carbohydrate metabolic process; the main CCs were the integral component of the membrane, nucleus, and membrane; and the main MFs were ATP binding, protein kinase activity, and oxidoreductase activity (Figure 5b). Additionally, in the C-S3 vs. D-S3 group, the main BPs were protein phosphorylation, carbohydrate metabolic process, photosynthesis, and response to abscisic acid; the main CCs were the integral component of the membrane, chloroplast, and photosystem I/II; and the main MFs were protein binding, ATP binding, and protein kinase activity (Figure 5c). Furthermore, in the C-S4 vs. D-S4 group, the main BPs were regulation of transcription, carbohydrate metabolic process, response to abscisic acid, and ethylene-activated signaling pathway; the main CCs were the integral component of the membrane and nucleus; and the main MFs were protein binding, DNA binding, and DNA-binding transcription factor activity (Figure 5d). Moreover, fifteen common GO terms, which were generally expressed in all the comparison groups, were found, such as carbohydrate metabolic process, response to abscisic acid, and ethylene-activated signaling pathway (Appendix A). These findings suggested that these pathways may play crucial roles in the spike development process.

### 2.7. KEGG Enrichment Analysis of DEGs

To further investigate the metabolic pathways involved in spike development, Kyoto Encyclopedia of Genes and Genomes (KEGG) pathway analysis was performed. We identified 99 unique KEGG enrichment pathways in the four different comparison groups (Appendix A). The top 20 KEGG pathways with the smallest significant *q*-values were selected from each comparison group (Figure 6). In the C-S1 vs. D-S1 group, the most enriched pathways were plant hormone signal transduction, starch and sucrose metabolism, MAPK signaling pathway-plant, photosynthetic antenna proteins, and carbon fixation in photosynthetic organisms (Figure 6a). In the C-S2 vs. D-S2 group, plant-pathogen interaction, MAPK signaling pathway-plant, plant hormone signal transduction, glycolysis/gluconeogenesis, and phenylpropanoid biosynthesis were significantly enriched (Figure 6b). Additionally, in the C-S3 vs. D-S3 group, plant hormone signal transduction, starch and sucrose metabolism, photosynthetic antenna proteins, and phenylpropanoid biosynthesis were significantly enriched (Figure 6c). Moreover, plant-pathogen interaction, starch and sucrose metabolism, plant hormone signal transduction, glyoxylate and dicarboxylate metabolism, and circadian rhythm-plant were significantly enriched in the C-S4 vs. D-S4 group (Figure 6d). Simultaneously, 31 common KEGG enrichment pathways were identified across the four different comparison groups, including plant hormone signal transduction, starch and sucrose metabolism, glyoxylate and dicarboxylate metabolism, and circadian rhythm-plant (Appendix A), which may be closely related to spike development.

### 2.8. Important Regulatory Pathways and DEGs Associated with Spike Development

Through the KEGG enrichment analysis, we found that many DEGs were involved in plant hormone signal transduction, starch and sucrose metabolism, photosynthetic antenna proteins, and circadian rhythm, which are considered important regulatory pathways related to spike development and flowering time. Therefore, we further analyzed the expression levels of genes associated with the four pathways in each comparison group.

Fifty-two DEGs involved in plant hormone signal transduction were identified in the four comparison groups (Appendix A). Most of the DEGs were related to auxin-responsive proteins (IAA and SAUR). Several DEGs associated with the abscisic acid receptor (PYL5 and PYL9), ABA protein phosphatase 2C (PP2C), ethylene receptor, and brassinosteroid LRR receptor kinase (BRI1) signaling pathways were also identified. Two genes associated with DELLA protein GAI were uniquely found and upregulated in the C-S1 vs. D-S1 group (Table 1). The results indicated that the expression of these genes may be one of the important reasons why line 18-1-5 headed earlier than its wheat parents. Additionally, we found that 144 DEGs were involved in the starch and sucrose metabolism pathway (Appendix A). Four DEGs were commonly shared in all comparison groups, among which one gene associated with sucrose synthase 1 (SUS1) was highly expressed. Likewise, seven genes associated with trehalose 6-phosphate phosphatase (TPP) RA3 were identified and upregulated in the C-S1 vs. D-S1 group, of which two were also identified in the C-S3 vs. D-S3 and C-S4 vs. D-S4 comparison groups, respectively, but their expression levels were downregulated.

Furthermore, we identified 53 DEGs in the photosynthetic antenna proteins pathway, which were mainly related to chlorophyll a/b-binding protein (Appendix A). Among these, 44 were identified and upregulated in the C-S1 vs. D-S1 group (Table 1), while 33 and 8 were found and downregulated in the C-S3 vs. D-S3 and C-S4 vs. D-S4 groups, respectively. Additionally, 47 DEGs related to the circadian rhythm pathway were identified (Appendix A), of which two genes related to two-component response regulator-like PRR37 were shared and downregulated in all the comparison groups. Three genes associated with the protein HEADING DATE 3A were exclusively identified and upregulated, while two related to the protein FLOWERING LOCUS T were downregulated in the C-S1 vs. D-S1 group. Similarly, three and one genes associated with the zinc finger protein CONSTANS-LIKE were uniquely identified and upregulated in the C-S1 vs. D-S1 and C-S4 vs. D-S4 groups, respectively, while two genes related to the zinc finger protein CONSTANS-LIKE were exclusively found and downregulated in the C-S3 vs. D-S3 group. Moreover, three genes related to cycling DOF factor 2 were also found and downregulated.

### 2.9. TFs Associated with Spike Development

Transcription factors (TFs) are involved in the regulation of plant development and flowering. In all libraries, we identified a total of 577 TFs, which were classified into 48 transcription factor families (Appendix A). Among these, 339, 114, 104, and 172 TFs were identified in the C-S1 vs. D-S1, C-S2 vs. D-S2, C-S3 vs. D-S3, and C-S4 vs. D-S4 comparison groups, respectively, and eight were common TFs. The key TFs associated with the plant flowering process were found, including *bHLH* (40 DEGs), *bZIP* (19 DEGs), *MADS-box* (49 DEGs), *MYB* (14 DEGs), *NAC* (38 DEGs), *SBP* (8 DEGs), *WRKY* (43 DEGs), and *NF-Y* (10 DEGs).

Among the *bHLH* gene family, we found most *bHLH* transcription factors to be significantly expressed in the four different developmental stages, of which *TabHLH25* was highly expressed in the C-S1 vs. D-S1 group, while *TabHLH93* was significantly downregulated in all comparison groups (Table 2). Among the *bZIP* gene family, the expression of *TaHY5* was upregulated in the C-S1 vs. D-S1 and C-S2 vs. D-S2 groups, and *TabZIP44* was highly expressed in the C-S4 vs. D-S4 group. For the MADS-box gene family, many known genes related to flowering time, including *TaFUL2*, *TaAGL6*, *TaSEP3*, *TaAG1*, and *TaSEP1-2*, were upregulated in the C-S1 vs. D-S1 group, as well as *TaVRT-2* was highly expressed in the C-S4 vs. D-S4 group. Interestingly, *TaSEP6* and *TaMADS32* were uniquely identified in the C-S1 vs. D-S1 group and expressed at high levels. Additionally, in the *MYB* gene family, the expression of *TaMYB94*, *TaMYB30*, and *TaMYB44* was significantly upregulated in the C-S3 vs. D-S3 and C-S4 vs. D-S4 groups. Similarly, in the *NAC* gene family, *TaNAC2* was highly expressed in the C-S1 vs. D-S1 group, and *TaNAC8* was always significantly upregulated across the four different developmental stages. All *SBP* transcription factor genes were downregulated in the comparison groups, such as *TaSPL17*. Among the *WRKY* gene family, *TaWRKY71* was exclusively identified and expressed at a high level in the C-S1 vs. D-S1 group. The expression of *TaWRKY46*, *TaWRKY11*, and *TaWRKY24* was upregulated in the C-S1 vs. D-S1, C-S2 vs. D-S2, and C-S4 vs. D-S4 groups, respectively. Among the *NF-Y* gene family, the expression of *TaNF-YB5* was highly expressed in the C-S1 vs. D-S1 group, and *TaNF-YC6* was upregulated in the C-S4 vs. D-S4 group.

### 2.10. Quantitative Real-Time PCR Validation of Spike Development-Related DEGs

To verify the reliability of our transcriptome data, we selected fifteen DEGs from circadian rhythm, photosynthetic antenna proteins, phytohormones, and transcription factors, and performed qRT-PCR analysis in all groups. Among these genes, ten were upregulated, three were downregulated, and two were up- or downregulated. The qRT-PCR expression patterns of the fifteen validated genes were found to follow the same trend as the RNA-seq results, further confirming the accuracy of the RNA-seq data acquired in this study (Figure 7).

## 3. Discussion

### 3.1. Wheat–Alien Species Are Potentially Valuable Germplasm Resources for Improving Early Heading and Maturity

Early heading and maturity play crucial roles in enhancing the multiple cropping index, optimizing land use efficiency, and maximizing final grain yield. Owing to these advantages, numerous early-maturing wheat varieties have been developed and released across various countries [38]. Nevertheless, the limited genetic diversity of cultivated wheat, resulting from the excessive reliance on early-maturing resources over time, has hindered advancements in grain yield improvements [39]. Numerous studies have demonstrated that transferring beneficial genes from wild relatives into wheat through distant hybridization is an effective strategy for improving wheat and increasing genetic diversity [40,41]. To date, several wheat early-maturing germplasms containing alien chromatin from wild relatives have been successfully developed, including wheat–rye 5R(5A) substitution lines [42], two wheat–barley disomic addition lines WB0528 and WB0647 [43], a wheat–barley 7H addition line [44], a wheat–*P. huashanica* 6Ns disomic addition line [45], and a wheat–*P. huashanica* 7Ns ditelosomic addition line [46]. In the present study, we found that the wheat–*P. huashanica* 7Ns disomic addition line 18-1-5 showed significantly faster spike development than its wheat parents from the double ridge stage, thereby resulting in 9–11 days earlier heading and 8–10 days earlier maturation (Figure 2 and Figure 3). Pedigree analysis revealed that chromosome 7Ns of *P. huashanica* contains genes that significantly accelerate heading and maturity in wheat. Previous studies have confirmed that the double ridge stage is a key point that determines flower induction and floral meristem development in wheat, thereby ensuring high and stable yields [26,47]. Accordingly, these findings suggest that early spike development, particularly during the double ridge stage, may be a significant factor contributing to earlier heading and maturation in line 18-1-5. This line represents a potentially valuable germplasm for breeding early-maturing wheat cultivars. Wheat heading time is primarily determined by *VRN*, *PPD*, and *Eps* genes [4,5]. However, the main factor controlling early heading in line 18-1-5 specifically still needs further research.

### 3.2. DEGs and Pathways Associated with Spike Development

In this study, we identified thousands of DEGs associated with spike development, most of which were differentially expressed at the double ridge stage (S1) (Figure 4), suggesting that this developmental period may be pivotal for the flowering initiation of 18-1-5. This result is similar to those reported by Feng et al. [48] and Liu et al. [25]. The GO analysis showed that pathways related to spike development, including carbohydrate metabolic process, photosynthesis, response to abscisic acid, and ethylene-activated signaling pathway, were significantly enriched across various comparison groups (Figure 5). The KEGG analysis revealed that the principal enrichment pathways included plant hormone signal transduction, starch and sucrose metabolism, photosynthetic antenna proteins, and circadian rhythm-plant (Figure 6). RNA-seq analysis using wheat *PHYTOCHROME B* and *PHYTOCHROME C* mutants identified many PHYB-regulated genes, which were mainly enriched in components of the auxin, gibberellin, and brassinosteroid biosynthesis and signaling pathways [49]. RNA-seq analysis at the double ridge stage and androgynous primordium differentiation stage showed that DEGs were mainly involved in carbohydrate metabolism, trehalose metabolic process, photosynthesis, light reaction, and hormone signaling [26]. A recent study revealed the different molecular mechanisms of flowering time between “Truman” and “Deguo 2” and found that a large number of DEGs were involved in circadian rhythm, plant hormone signaling, phenolamides, and antioxidants [50]. Our findings show slight deviations from previous studies, underscoring the complexity inherent within the regulatory network governing flowering time. Overall, these results suggest that DEGs within these critical pathways play potentially significant roles in wheat spike development processes.

### 3.3. Plant Hormone Signal Transduction and Starch and Sucrose Metabolism-Related Genes Involved in Spike Development

Plant hormones, such as IAA, ABA, BR, GA, MeJA, and ethylene, play a vital role in regulating plant flowering time by coordinating various signal transduction pathways. For example, *AUXIN RESPONSE FACTOR4* (*FaARF4*) promotes flowering by activating the floral meristem identity genes *APETALA1* (*AP1*) and *FRUITFULL* (*FUL*) in woodland strawberry [51]. The overexpression of the AUX/IAA gene *TaIAA15* causes early flowering time by interacting with the auxin response factor (*ARF*) in wheat [52]. The loss-of-function of the auxin-responsive gene *OsSAUR56* results in an early heading date in rice [53]. Moreover, BRI1-EMS-SUPPRESSOR 1 (BES1), a key regulator in the BR pathway, positively regulates photoperiodic flowering in *Arabidopsis* through the BES1-BEE1-FT signaling pathway [54]. DELLA degradation by GA promotes flowering through the GAF1-TPR-dependent repression of floral repressors in *Arabidopsis* [55]. GA signaling regulates flowering via the DELLA–BRAHMA–NF-YC module in *Arabidopsis* [56]. In *Arabidopsis*, ethylene can accelerate the transition from vegetative growth to flowering [57]. Some ethylene receptor genes (*AcERS1b*, *AcETR2a*, and *AcETR2b*) play important roles in affecting pineapple flowering [58]. In this research, we found that many DEGs associated with the hormone signal transduction pathway, such as *ARF*, *AUX/IAA*, *SAUR*, *DELLA*, *BRI1*, and *ETR*, were differentially expressed at different developmental stages (Table 1; Appendix A), indicating that the flowering time of 18-1-5 may be related to changes in endogenous hormone levels caused by the upregulation/downregulation of these genes.

Previous studies reported that starch and sucrose participate in the regulation of plant flowering transition. The starch content in both leaves and buds increases during the flower induction process [59]. The expression level of the key starch synthase gene *GBSSI* related to starch deposition can promote plant flowering by increasing the *CO* expression [60]. The trehalose precursor trehalose-6-phosphate (T6P), dephosphorylated by trehalose-6-phosphate phosphatase (TPP), is suggested to function as a proxy for carbohydrate status in plants, and the loss of the gene encoding T6P synthase 1 (*TPS1*) can cause *Arabidopsis* to flower extremely late [61]. The overexpression of the gene encoding an O-linked N-acetylglucosamine (O-GlcNAc) transferase (*TaOGT1*) accelerates heading date in winter wheat, which is mainly associated with sugar content and the transcript levels of flowering time genes [62]. In our study, one gene linked to sucrose synthase 1 (SUS1) was highly expressed in all comparison groups, and seven genes associated with TPP were upregulated in the C-S1 vs. D-S1 group (Table 1). These results suggested that the expression of genes related to SUS1 and TPP may promote spike development and early flowering of 18-1-5, and that TPP-related genes may mainly function at the double-ridge stage.

### 3.4. Photosynthetic Antenna Proteins and Circadian Rhythm-Related Genes Involved in Spike Development

The antenna system is an important regulator in PSI and PSII for photosynthesis [63]. Light-harvesting chlorophyll a/b-binding (LHC) proteins, also known as antenna proteins, play essential roles in absorbing light and transferring energy for plant growth and development [64]. In previous studies, *Zm00001d009589* (*lhcb3*) was reported to be involved in chloroplast development [65]. RNA-seq analysis revealed many DEGs related to photosynthetic antenna proteins in response to BR signaling in maize [66]. In this study, we found that 44 DEGs related to chlorophyll a/b-binding protein were identified and upregulated in the C-S1 vs. D-S1 group (Table 1; Appendix A), suggesting that the upregulated expression of these genes at the double ridge stage (S1) may be important for accelerating the early development and flowering of 18-1-5.

Circadian rhythm plays a pivotal role in the regulation of plant flowering. The pseudoresponse regulator protein 37 (*PRR37*), which is regulated by the circadian clock, modulates flowering time by activating the expression of the floral inhibitor *CO* and repressing the expression of the floral activators *Early Heading Date 1* (*Ehd1*) and *FT* in sorghum [67,68]. *Heading date 3a (Hd3a)*, a rice ortholog of the *Arabidopsis FT* gene, is upregulated by *Hd1*, a homolog of *CO*, and promotes heading time under SD conditions [69,70]. *Rice Flowering Locus T 1* (*RFT1*) regulates heading date and influences yield traits in rice [71]. The overexpression of the CONSTANS (CO)-like protein *OsCOL15* results in a delayed flowering phenotype by promoting the flowering repressor Grain number, plant height, and heading date 7 (Ghd7) and repressing the flowering activator Rice Indeterminate 1 (RID1) in rice [72]. *CO-like 9* (*OsCOL9*) can delay flowering time in rice by repressing the *Ehd1* pathway [73]. *LATE BLOOMER2* (*LATE2*) is a cycling DOF factor (CDF) homolog that can regulate *FT* expression and flowering time without affecting the expression of *CO-like* genes [74]. In this study, we found that several genes described as two-component response regulator-like PRR37, protein HEADING DATE 3A, protein FLOWERING LOCUS T, zinc finger protein CONSTANS-LIKE, and cycling DOF factor 2 were differentially expressed at the different developmental stages (Table 1), indicating that the upregulation/downregulation of these genes may be involved in the spike development process for 18-1-5.

### 3.5. TFs Involved in Spike Development

TFs, such as *bHLH*, *bZIP*, *MADS-box*, *MYB*, *NAC*, *SBP*, *WRKY*, and *NF-Y*, play important roles in the regulation of plant development and flowering. For example, *AtbHLH93* has been shown to promote flowering under SD conditions by repressing the floral repressor *MAF5* in *Arabidopsis* [75]. NO FLOWERING IN SHORT DAY (NFL), a bHLH transcription factor, promotes flowering specifically under SD conditions through the GA signaling pathway in *Arabidopsis* [76]. In this study, *TabHLH25* was highly expressed in the C-S1 vs. D-S1 group, and *TabHLH93* was significantly downregulated in all comparison groups (Table 2). The results suggested that *TabHLH25* and *TabHLH93* may be potentially important for affecting the spike development of 18-1-5. Reports have shown that the bZIP transcription factor ELONGATED HYPOCOTYL 5 (HY5), a positive regulator of light signaling, promotes photomorphogenesis by directly or indirectly interacting with several other downstream factors and controls almost one-third of the gene expression of the *Arabidopsis* genome [77]. In this study, we discovered that *TaHY5* was highly expressed in the C-S1 vs. D-S1 and C-S2 vs. D-S2 groups (Table 2), speculating that this gene might indirectly affect wheat heading time by interacting with other factors. Many studies have demonstrated that MADS-box genes encoding a family of transcription factors control flowering time and diverse developmental processes in plants [78,79]. In this research, we found that several known MADS-box genes, including *TaFUL2*, *TaAGL6*, *TaSEP3*, *TaAG1*, *TaVRT2*, and *TaSEP1-2*, were significantly upregulated (Appendix A), suggesting that these genes play important roles in regulating spike development for 18-1-5. Specifically, two MADS-box transcription factor genes, *TaSEP6* and *TaMADS32*, were highly expressed in the C-S1 vs. D-S1 group (Table 2). *SEPALLATA* (*SEP*)*-like* genes participate in every step of reproductive growth, ranging from the initiation of inflorescence meristems to the determination of floral organs [80]. *OsMADS32* has been proven to interact with PISTILLATA (PI)-like proteins and regulate flower development in rice [81]. Therefore, we speculated that *TaSEP6* and *TaMADS32* might be essential in regulating the spike development of 18-1-5.

Increased levels of *MYB30* could accelerate flowering by interacting with the *FT* promoter in *Arabidopsis* [82]. The transcription factor *TaMYB72* not only promotes flowering in rice but also directly activates the expression of *TaFT*, thereby promoting heading and enhancing grain yield traits in wheat [83,84]. In our study, *TaMYB94*, *TaMYB30*, and *TaMYB44* were significantly upregulated in the C-S3 vs. D-S3 and C-S4 vs. D-S4 groups, suggesting that they may affect the flowering time of 18-1-5. Additionally, we also found that *TaNAC2* was highly expressed in the C-S1 vs. D-S1 group (Table 2). The *Picea wilsonii* transcription factor *PwNAC2* has been proven to interact with the Resemble-FCA-contain-PAT1 domain (PwRFCP1) to participate in flowering regulation [85]. Thus, we speculated that *TaNAC2* may take part in flowering induction for 18-1-5. Our study found that the SBP transcription factor gene *TaSPL17* was downregulated in the C-S2 vs. D-S2 group. Its orthologous gene in *Arabidopsis* is *ATSPL9*, which can regulate flowering time by promoting the transcription of *FUL*, *SOC1*, and *AGL42* [86,87]. Thus, we deduced that the transcription of *TaSPL17* may be suppressed during spike development, thereby affecting *TaSPL17* activity and heading time. In this research, the expression of *TaWRKY71* was significantly upregulated in the C-S1 vs. D-S1 group. *AtWRKY71* accelerates flowering via the direct activation of *FT* and *LEAFY (LFY)* in *Arabidopsis* [88]. Thus, we hypothesized that *TaWRKY71* may promote flowering for 18-1-5. Previous studies have shown that NF-YB and NF-YC genes can regulate flowering in plants [89,90]. Our study found that *TaNF-YB5* and *TaNF-YC6* were significantly upregulated (Table 2), suggesting that these two genes may also be involved in the heading time of 18-1-5. In summary, the key TFs identified in this study are pivotal for the regulation of spike development and flowering time in 18-1-5, but their specific regulatory mechanism needs further exploration.

## 4. Materials and Methods

### 4.1. Plant Materials

The plant materials used in the current study included common wheat *Triticum aestivum* cv. Chinese Spring (CS, 2*n* = 6*x* = 42, AABBDD, from Sichuan province, China), the Chinese Spring *ph2b* (CS*ph2b*) mutant, and the wheat–*P. huashanica* 7Ns disomic addition line 18-1-5 (2*n* = 44 = 42W + II7Ns). The 7Ns disomic addition line 18-1-5 was developed and identified from the BC_1_F_5_ generation of CS*ph2b*/*P. huashanica* accession ZY3156 (2*n* = 2*x* = 14, NsNs, from Shaanxi province, China)//CS///CS [35]. All the materials were preserved at the Triticeae Research Institute, Sichuan Agricultural University, China.

### 4.2. GISH and FISH Analyses

Chromosome preparation of root-tip cells at mitotic metaphase was conducted and subjected to sequential GISH and FISH analyses [35,91]. The total genomic DNA of *P. huashanica* was labeled with dUTP-ATO-550 (Jena Bioscience, Jena, Germany) and used as a probe for GISH, and CS genomic DNA was included as blocking DNA. Five oligonucleotide probes, Oligo-pSc119.2, Oligo-pTa535, Oligo-pSc200, Oligo-44, and Oligo-pTa71A-2, were labeled with TAMRA-5′ or 6-FAM-5′ for FISH [36,37]. An Olympus BX63 fluorescence microscope equipped with a Photometric SenSys DP-70 CCD camera (Olympus Corporation, Tokyo, Japan) was used to capture the chromosome fluorescent signals.

### 4.3. Phenotypic Characterization

The seeds of CS, CS*ph2b*, and 18-1-5 were planted in a randomized complete block design with three replicates in the fields at the Wenjiang experimental field (Chengdu, China) during three consecutive growing seasons from 2020 to 2023. The field layout consisted of 50 rows for each material, with each row being 1.5 m in length and 0.3 m in spacing, with 15 grains per row. Spike differentiation, including apex elongation, single-ridge, double-ridge, glume primordia differentiation, floret primordia differentiation, stamen and pistil differentiation, anther separation, and tetrad stages, was investigated for the main stems of ten randomly selected plants at seven-day intervals, using a stereomicroscope (ZEISS SteREO Discovery.V20) following the wheat spike differentiation criteria [92]. The heading time was recorded as days from the sowing date to the date when approximately 50% of spikes fully emerged from the flag leaf sheath. The maturity time was calculated from the sowing date to the date when their endosperm turned waxy and the grains had a 40% moisture content [93]. Statistical analysis of phenotypic data was performed using SPSS version 25.0 (IBM Corp., Armonk, NY, USA). Student’s *t*-test was conducted to determine the significance level of phenotypic differences between 18-1-5 and its wheat parents.

### 4.4. RNA Extraction, Library Construction, and Illumina Sequencing

Young spikes from the main stems of each material at four different developmental stages, namely, the double-ridge stage (S1), the glume primordia differentiation stage (S2), the floret primordia differentiation stage (S3), and the stamen and pistil differentiation stage (S4), were collected. Approximately 40–50 young spikes at each developmental stage were pooled with three biological replicates. The collected samples were immediately frozen in liquid nitrogen and stored at −80 °C for subsequent RNA extraction. Total RNA was isolated using Trizol reagent (Invitrogen, Carlsbad, CA, USA) according to the manufacturer’s procedure. RNA concentration was quantified using a NanoDrop 2000 spectrophotometer (Thermo Fisher Scientific, Wilmington, DE, USA). RNA purity and integrity were checked using a 2100 Bioanalyzer RNA 6000 Nano LabChip Kit (Agilent, Santa Clara, CA, USA). High-quality RNA samples with RIN number > 7.0 were used to construct the sequencing library.

The mRNA was purified from total RNA (5 μg) using Dynabeads Oligo (dT) (Thermo Fisher, CA, USA) with two rounds of purification and subjected to RNA fragmentation. Afterward, the cleaved RNA fragments were reverse-transcribed to create the first-strand cDNA by SuperScript™ II Reverse Transcriptase (Invitrogen, USA), followed by the synthesis of the second-strand cDNA. The cDNA library was constructed by PCR amplification, and the average insert size for the final cDNA libraries was 300 ± 50 bp [94]. Finally, the 2 × 150 bp paired-end sequencing (PE150) was performed on an Illumina NovaSeq™ 6000 platform (LC-Bio Technology Co., Ltd., Hangzhou, China) following the vendor’s recommended protocol.

### 4.5. Transcriptome Analysis

Raw reads in FASTQ format were first processed using in-house Perl scripts. The poly-N, adapters, and low-quality reads from raw data were removed to obtain high-quality clean reads for all downstream analyses, and the sequence quality of clean data, including Q20, Q30, and GC-content, was verified using FastQC (http://www.bioinformatics.babraham.ac.uk/projects/fastqc/, accessed on 10 September 2024). After that, the clean reads of all samples were mapped to the CS reference genome (IWGSC Refseq v1.1, https://wheat-urgi.versailles.inra.fr/Seq-Repository/Annotations, accessed on 5 June 2025) using the HISAT2 v2.2.0 package [95]. StringTie v2.1.2 (http://ccb.jhu.edu/software/stringtie/, accessed on 20 September 2024) [96] and GffCompare (http://ccb.jhu.edu/software/stringtie/gffcompare.shtml, accessed on 28 September 2024) software were used to assemble the transcripts and reconstruct a comprehensive transcriptome, respectively.

The fragment per kilobase of transcript per million mapped reads (FPKM) value was calculated to estimate the expression level of genes in each sample. Differential gene expression analysis was performed using DESeq2 v1.26.0 software between two different groups [97]. The genes with the parameter of false discovery rate (FDR) < 0.05 and |log_2_(foldchange)| ≥ 1 were considered as differentially expressed genes (DEGs). Gene functions were annotated using the Nr, Swiss-Prot, COG, KEGG, GO, KOG, Pfam, and eggNOG databases. Significant GO and KEGG enrichment analyses for DEGs were performed using the OmicStudio tools (https://www.omicstudio.cn/tool, accessed on 25 June 2025) with a Q-value less than 0.05.

### 4.6. Transcription Factor Analysis

To identify transcription factor (TF) changes during spike development, the nucleotide sequences of DEGs from IWGSC RefSeq v1.1 were extracted and subjected to the identification and classification of TFs using the online tool iTAK (Plant Transcription factor & Protein Kinase Identifier and Classifier, http://itak.feilab.net/cgi-bin/itak/index.cgi, accessed on 5 June 2025) [98].

### 4.7. qRT-PCR Analysis

To validate the RNA-seq results, fifteen DEGs obtained from the sequencing data were selected for qRT-PCR analysis. Total RNA extraction from the collected samples was conducted as described above. Gene-specific primers were designed with Primer Premier 5 software (Premier Biosoft, Palo Alto, CA, USA) (Appendix A). cDNA was synthesized from 1.5 μg of DNase-treated total RNA using the Thermo RevertAid First Strand cDNA Synthesis Kit (Thermo-Fisher Scientific, Shanghai, China). The qRT-PCR was performed using the SYBR Premix pro Taq HS qPCR Kit (Accurate Bio Co., Ltd., Changsha, Hunan, China) following the manufacturer’s recommendations on the CFX96 Real-Time PCR System (Bio-Rad, Hercules, CA, USA) as described previously [99]. Each experiment included three technical replicates and at least three biological replicates. The wheat *Actin* gene was used as the internal reference gene for normalization. Relative expression levels were calculated using the 2^−ΔΔCt^ method [100].

## 5. Conclusions

In the present study, we found that the spike development of the wheat–*P. huashanica* 7Ns disomic addition line 18-1-5, particularly from the double ridge stage, was distinctly faster than that of its wheat parents, thereby leading to early heading and maturation. The transcriptome analysis of four different spike development stages revealed several key genes involved in plant hormone signal transduction, starch and sucrose metabolism, photosynthetic antenna proteins, and circadian rhythm, which were differentially expressed at different developmental stages, especially at the double ridge stage, implying their influence on the spike development and flowering time. Additionally, a large number of differentially expressed TFs, including bHLH, bZIP, MADS-box, NAC, WRKY, SBP, NF-Y, and MYB gene families, were also found, but only a few TFs, such as *TabHLH93*, *TaHY5*, *TaSEP6*, *TaSPL17*, *TaMYB30*, *TaNAC2*, *TaNF-YB5*, and *TaWRKY71*, were significantly up- or downregulated, suggesting that they might play important roles in affecting the spike development and flowering time of 18-1-5. Our results provide useful information for future investigations into the molecular mechanisms of wheat heading, such as gene function validation and protein–protein interactions.

## Figures and Tables

**Figure 1 plants-14-02077-f001:**
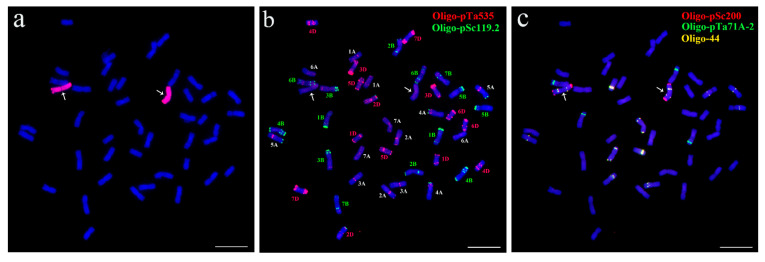
GISH and FISH identification of wheat–*P. huashanica* 7Ns disomic addition line 18-1-5. (**a**) *P. huashanica* genomic DNA labeled in red as a GISH probe. (**b**,**c**) Oligo-pTa535 (red), Oligo-pSc119.2 (green), Oligo-pSc200 (red), Oligo-pTa71A-2 (green), and Oligo-44 (yellow) as FISH probes. Arrows indicate *P. huashanica* chromosome 7Ns in 18-1-5. Scale bars: 10 μm.

**Figure 2 plants-14-02077-f002:**
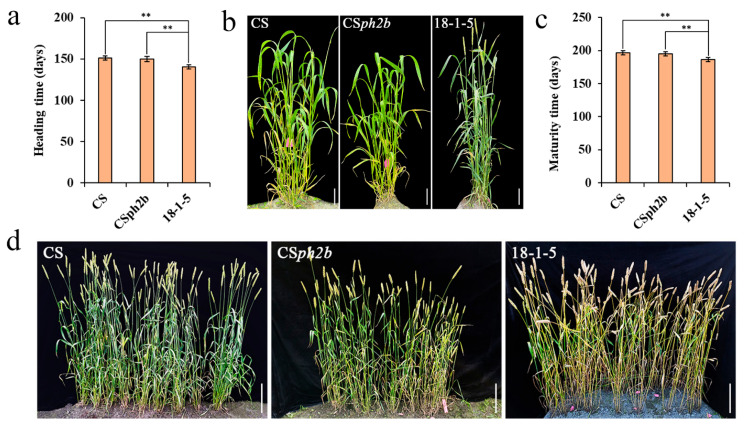
Investigation of heading and maturity times for wheat–*P. huashanica* 7Ns disomic addition line 18-1-5 and its wheat parents. (**a**,**b**) Statistical analysis and phenotype visualization of heading time under field conditions. (**c**,**d**) Statistical analysis and phenotype visualization of maturity time under field conditions. ** *p* < 0.01, two-tailed *t*-test. Error bars represent the standard deviation. Scale bars: 30 cm.

**Figure 3 plants-14-02077-f003:**
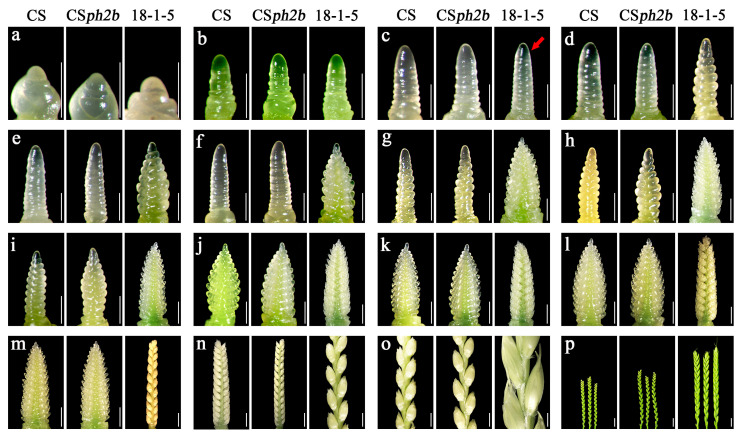
Dynamic observations of spike differentiation for 18-1-5 and its wheat parents under field conditions. (**a**–**p**) Microscope visualization showing the different spike development stages at 24, 34, 43, 53, 60, 67, 74, 82, 89, 96, 102, 110, 118, 126, 135, and 143 days after sowing, respectively. The sowing date for all the materials was 21 October 2022. The red arrow indicates the differences in spike development between 18-1-5 and its wheat parents. Scale bars: (**a**–**l**) 0.5 mm; (**m**–**o**) 2 mm; (**p**) 0.5 cm.

**Figure 4 plants-14-02077-f004:**
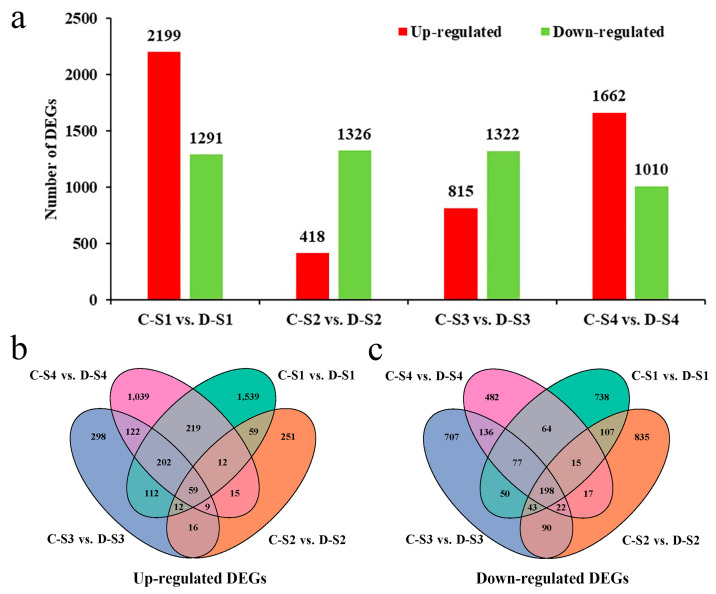
Distribution of DEGs at four different developmental stages. (**a**) The number of DEGs in each comparison group. (**b**) Venn diagram of upregulated DEGs in each comparison group. (**c**) Venn diagram of downregulated DEGs in each comparison group. The capital letter “C” represents 18-1-5, and “D” indicates CS and CS*ph2b*. S1, S2, S3, and S4 represent the double-ridge stage, the glume primordia differentiation stage, the floret primordia differentiation stage, and the stamen and pistil differentiation stage, respectively.

**Figure 5 plants-14-02077-f005:**
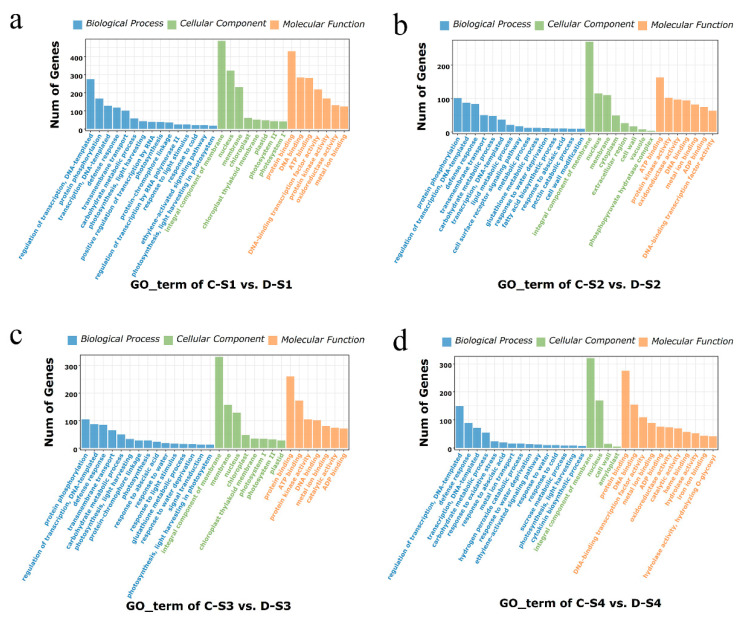
GO enrichment analysis of DEGs in each comparison group. (**a**) The most significantly enriched GO terms of C-S1 vs. D-S1. (**b**) The most significantly enriched GO terms of C-S2 vs. D-S2. (**c**) The most significantly enriched GO terms of C-S3 vs. D-S3. (**d**) The most significantly enriched GO terms of C-S4 vs. D-S4. The capital letter “C” represents 18-1-5, and “D” indicates CS and CS*ph2b*. S1, S2, S3, and S4 represent the double-ridge stage, the glume primordia differentiation stage, the floret primordia differentiation stage, and the stamen and pistil differentiation stage, respectively.

**Figure 6 plants-14-02077-f006:**
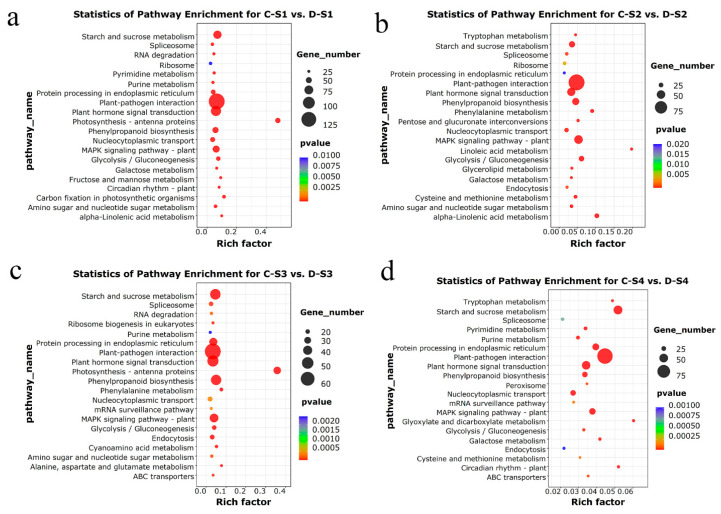
KEGG enrichment analysis of DEGs in each comparison group. (**a**) Top 20 KEGG enrichment scatter plot of DEGs in C-S1 vs. D-S1. (**b**) Top 20 KEGG enrichment scatter plot of DEGs in C-S2 vs. D-S2. (**c**) Top 20 KEGG enrichment scatter plot of DEGs in C-S3 vs. D-S3. (**d**) Top 20 KEGG enrichment scatter plot of DEGs in C-S4 vs. D-S4. The capital letter “C” represents 18-1-5, and “D” indicates CS and CS*ph2b*. S1, S2, S3, and S4 represent the double-ridge stage, the glume primordia differentiation stage, the floret primordia differentiation stage, and the stamen and pistil differentiation stage, respectively.

**Figure 7 plants-14-02077-f007:**
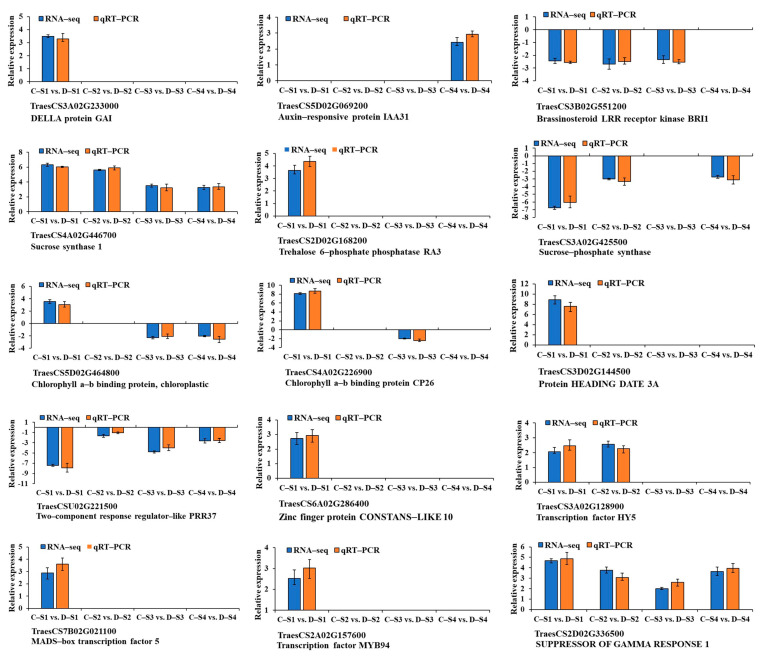
The expression level validation of fifteen DEGs using qRT-PCR. An empty histogram indicates that either RNA-seq could not detect the gene at this stage or that its expression level was low, with |log2(foldchange)| < 1. The capital letter “C” represents 18-1-5, and “D” indicates CS and CS*ph2b*. S1, S2, S3, and S4 represent the double-ridge stage, the glume primordia differentiation stage, the floret primordia differentiation stage, and the stamen and pistil differentiation stage, respectively. Error bars represent the standard deviation.

**Table 1 plants-14-02077-t001:** DEGs related to spike development in important regulatory pathways.

KEGG Pathways	Gene ID	Functional Annotation	C1 vs. D1	C2 vs. D2	C3 vs. D3	C4 vs. D4
Plant hormone signal transduction	TraesCS3A02G233000	DELLA protein GAI	3.04 ↑	-	-	-
	TraesCS3D02G220100	DELLA protein GAI	3.21 ↑	-	-	-
Starch and sucrose metabolism	TraesCS4A02G446700	Sucrose synthase 1 (SUS1)	6.33 ↑	5.62 ↑	3.53 ↑	3.27 ↑
	TraesCS2A02G161000	Trehalose 6-phosphate phosphatase RA3	10.63 ↑	-	-	-
	TraesCS2A02G161100	Trehalose 6-phosphate phosphatase RA3	22.77 ↑	-	-	-
	TraesCS2A02G161200	Trehalose 6-phosphate phosphatase RA3	7.00 ↑	-	2.26 ↓	-
	TraesCS2B02G187100	Trehalose 6-phosphate phosphatase RA3	3.17 ↑	-	-	-
	TraesCS2B02G187200	Trehalose 6-phosphate phosphatase RA3	5.71 ↑	-	-	-
	TraesCS2D02G168100	Trehalose 6-phosphate phosphatase RA3	10.49 ↑	-	-	2.29 ↓
	TraesCS2D02G168200	Trehalose 6-phosphate phosphatase RA3	3.66 ↑	-	-	-
Photosynthetic antenna proteins	TraesCS7A02G227100	Chlorophyll a/b-binding protein 1B-21, chloroplastic	3.90 ↑	-	-	-
	TraesCS7B02G192500	Chlorophyll a/b-binding protein 1B-21, chloroplastic	5.49 ↑	-	-	-
	TraesCS7D02G227300	Chlorophyll a/b-binding protein 1B-21, chloroplastic	4.81 ↑	-	-	-
Circadian rhythm—plant	TraesCSU02G199500	Two-component response regulator-like PRR37	37.05 ↑	8.16 ↑	69.31 ↑	4.75 ↑
	TraesCSU02G221500	Two-component response regulator-like PRR37	47.40 ↑	7.29 ↑	73.61 ↑	7.61 ↑
	TraesCS3A02G143100	Protein HEADING DATE 3A (Hd3a)	4.77 ↑	-	-	-
	TraesCS3B02G162000	Protein HEADING DATE 3A (Hd3a)	13.85 ↑	-	-	-
	TraesCS3D02G144500	Protein HEADING DATE 3A (Hd3a)	8.87 ↑	-	-	-
	TraesCS2B02G365300	Protein FLOWERING LOCUS T	4.98 ↓	-	-	-
	TraesCS5A02G297300	Protein FLOWERING LOCUS T	5.41 ↓	-	-	-
	TraesCS6A02G286400	Zinc finger protein CONSTANS-LIKE 10	2.73 ↑	-	-	-
	TraesCS4B02G045700	Zinc finger protein CONSTANS-LIKE 16	7.03 ↑	-	-	-
	TraesCS7B02G113400	Zinc finger protein CONSTANS-LIKE 16	2.23 ↑	-	-	-
	TraesCS7D02G209000	Zinc finger protein CONSTANS-LIKE 16	-	-	-	3.42 ↓
	TraesCS6D02G274100	Zinc finger protein CONSTANS-LIKE 16	-	-	2.31 ↓	-
	TraesCS2D02G351900	Zinc finger protein CONSTANS-LIKE 5	-	-	5.67 ↓	-
	TraesCS3D02G185500	Cycling DOF factor 2	2.52 ↓	-	-	-
	TraesCS3A02G180600	Cycling DOF factor 2	-	-	2.08 ↓	-
	TraesCS3B02G210300	Cycling DOF factor 2	-	-	-	2.27 ↓

The values represent the FPKM multiple of up- or downregulated DEGs. “-” indicates that either RNA-seq could not detect the gene at this stage or that its expression level was low, with |log2(foldchange)| < 1. The arrows “↑” and “↓” indicate gene up-regulation and down-regulation, respectively. The capital letter “C” represents 18-1-5, and “D” indicates CS and CS*ph2b*.

**Table 2 plants-14-02077-t002:** The key transcription factor genes associated with spike development.

TFs Family	Gene Name	Gene ID	C1 vs. D1	C2 vs. D2	C3 vs. D3	C4 vs. D4
bHLH	*TabHLH25-5B*	TraesCS5B02G518400	3.23 ↑	-	-	-
	*TabHLH25*	TraesCSU02G075200	3.99 ↑	2.97 ↓	-	-
	*TabHLH93*	TraesCS7A02G543300	32.90 ↓	10.54 ↓	50.51 ↓	111.88 ↓
bZIP	*TaHY5*	TraesCS3A02G128900	2.07 ↑	2.57 ↑	-	-
	*TabZIP44*	TraesCS6B02G124700	-	3.29 ↓	-	4.02 ↑
MADS-box	*TaSEP6-7A*	TraesCS7A02G122000	31.03 ↑	-	-	-
	*TaSEP6-7B*	TraesCS7B02G020800	17.46 ↑	-	-	-
	*TaSEP6-7D*	TraesCS7D02G120500	8.30 ↑	-	-	-
	*TaMADS32-3A*	TraesCS3A02G284400	2.18 ↑	-	-	-
	*TaMADS32-3B*	TraesCS3B02G318300	2.36 ↑	-	-	-
	*TaMADS32-3D*	TraesCS3D02G284200	2.61 ↑	-	-	-
MYB	*TaMYB94*	TraesCS2A02G157600	-	-	2.53 ↑	2.76 ↑
	*TaMYB30*	TraesCS2B02G183100	-	-	2.16 ↑	2.42 ↑
	*TaMYB44*	TraesCS6B02G201700	-	-	3.15 ↑	3.63 ↑
NAC	*TaNAC2-5A*	TraesCS5A02G468300	6.48 ↑	-	-	-
	*TaNAC2-5B*	TraesCS5B02G480900	10.83 ↑	-	-	-
	*TaNAC8*	TraesCS2D02G336500	2.66 ↑	1.52 ↑	1.05 ↑	3.66 ↑
SBP	*TaSPL17-7A*	TraesCS7A02G246500	-	2.48 ↑	-	-
	*TaSPL17-7B*	TraesCS7B02G144900	-	2.51 ↑	-	-
	*TaSPL17-7D*	TraesCS7D02G245200	-	2.83 ↑	-	-
WRKY	*TaWRKY71-6A*	TraesCS6A02G146900	4.28 ↑	-	-	-
	*TaWRKY71-6B*	TraesCS6B02G175100	5.86 ↑	-	-	-
	*TaWRKY71-6D*	TraesCS6D02G136200	5.94 ↑	-	-	-
	*TaWRKY46*	TraesCS5B02G183800	2.80 ↑	-	-	-
	*TaWRKY11*	TraesCS2D02G431000	-	2.31 ↑	-	-
	*TaWRKY24*	TraesCS3B02G379200	-	-	-	2.49 ↑
NF-Y	*TaNF-YB5-3A*	TraesCS3A02G457100	8.09 ↑	-	-	-
	*TaNF-YB5-3D*	TraesCS3D02G450100	24.10 ↑	-	-	-
	*TaNF-YC6*	TraesCS5D02G265000	-	-	-	2.96 ↑

The values represent the FPKM multiple of up- or downregulated DEGs. “-” indicates that either RNA-seq could not detect the gene at this stage or that its expression level was low, with |log2(foldchange)| < 1. The arrows “↑” and “↓” indicate gene up-regulation and down-regulation, respectively. The capital letter “C” represents 18-1-5, and “D” indicates CS and CS*ph2b*.

## Data Availability

The datasets generated and analyzed during the current study are available from the National Genomics Data Center (NGDC) under BioProject ID: PRJCA030627 (https://ngdc.cncb.ac.cn/).

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
