# Peer review of "Transcriptome Profiling of Spike Development Reveals Key Genes and Pathways Associated with Early Heading in Wheat–Psathyrstachys huashanica 7Ns Chromosome Addition Line"

_plants, 2025, doi:10.3390/plants14132077_

Round 1

Reviewer 1 Report

Comments and Suggestions for Authors

The authors developed that a cytogenetically stable wheat–P. huashanica 7Ns disomic chromosome addition line (2n = 44 = 42W + â…¡7Ns) and revealed it sows early-heading compared with original wheat line CS. This is a highly commendable study and interesting research material.

Why is this chromosome-added line early in life? In trying to figure it out, the authors have a major misconception.

The earliness of wheat is mainly determined by three factors: photoperiod responsiveness, vernalization requirement, and non-environmental early-late trait.

The author first needs to clarify where the chromosome addition lines differ from the original variety in these factors.

The authors performed transcriptome analysis at four different developmental stages of the 7Ns disomic addition line and its wheat parents. A total of 10,043 differentially expressed genes (DEGs) were identified during spike development. Herein lies the author's major misconception.

The authors are looking for differences in gene patterns involved in the ear formation process, not differences in gene patterns related to early-heading.

The determination of the timing of flowering/heading time is made at the end of the vegetative growth period. The time young spike primordia are formed already entered the reproductive growth phase. Therefore analysis of gene expression patterns at that time does not reveal the early-heading mechanism. 

The authors need to analyze gene expression patterns during the transition in leaves from the nutritional growth phase to the reproductive growth phase.

Author Response

Dear Reviewer 1:

Thank you very much for taking the time to review our manuscript entitled ‘Transcriptome profiling reveals key genes and pathways associated with early heading in wheat–Psathyrostachys huashanica 7Ns chromosome addition line’ (ID: plants-3681655). Those comments are all valuable and very helpful for revising and improving our paper, as well as the important guiding significance to our researches. We have studied these comments carefully and made the necessary corrections, which we hope will meet with your approval. Revised portion are marked up using the “Track Changes” function in the manuscript. The main corrections in the manuscript and the point-by-point responses to your comments are as follows.

Sincerely yours,

Houyang Kang

Point-by-point response to Reviewer 1

Comments 1: The authors developed that a cytogenetically stable wheat–P. huashanica 7Ns disomic chromosome addition line (2n = 44 = 42W + â…¡7Ns) and revealed it sows early-heading compared with original wheat line CS. This is a highly commendable study and interesting research material.

Response 1: Thank you very much for your valuable comments. We have considered these comments carefully and tried our best to address every one of them.

Comments 2: Why is this chromosome-added line early in life? In trying to figure it out, the authors have a major misconception.

Response 2: Many thanks for your insightful comments. Previous studies have shown that P. huashanica exhibits earlier heading and maturation characteristics compared to other Triticeae species (Chen et al. 1991; Fu et al. 2003; Kang et al. 2008). In this study, the line 18-1-5, which carries an additional pair of 7Ns chromosomes, was derived from the BC1F5 generation of CSph2b/P. huashanica//CS///CS. Based on pedigree and phenotypic data analysis, we hypothesize that chromosome 7Ns of P. huashanica harbors alien genes that significantly accelerate heading and maturity in wheat, thereby resulting in a shorter growth period for line 18-1-5 compared to its wheat parental lines. We added the following information in the revised manuscript to make this clear in lines 394-395: ‘Pedigree analysis revealed that chromosome 7Ns of P. huashanica contains genes that significantly accelerate heading and maturity in wheat.’

Comments 3: The earliness of wheat is mainly determined by three factors: photoperiod responsiveness, vernalization requirement, and non-environmental early-late trait. The author first needs to clarify where the chromosome addition lines differ from the original variety in these factors.

Response 3: Thank you for your comments. Through three consecutive years of field investigation, our results confirmed that the addition line 18-1-5 exhibited earlier heading and maturity compared to its wheat parental lines. This suggests that chromosome 7Ns of P. huashanica harbors genes that significantly accelerate heading and maturity in wheat, which differs from the genetic characteristics of the original wheat varieties. To date, we have conducted a pre-experiment to explore these factors influencing the early heading of the addition line 18-1-5, and initially found that the early heading trait for 18-1-5 was mainly controlled by non-environmental early-late trait (i.e., earliness per se). However, it still needs to be verified by multiple field locations and years data in future research. We added the following information in the revised manuscript in lines 401-403: ‘Wheat heading time is primarily determined by VRNPPD, and Eps genes [4, 5]. However, the main factor controlling early heading in line 18-1-5 specifically still requires further research.’

Comments 4: The authors performed transcriptome analysis at four different developmental stages of the 7Ns disomic addition line and its wheat parents. A total of 10,043 differentially expressed genes (DEGs) were identified during spike development. Herein lies the author's major misconception. The authors are looking for differences in gene patterns involved in the ear formation process, not differences in gene patterns related to early-heading.

Response 4: Many thanks for your comments. We changed some statements for ‘heading time’ to ‘spike development’ in the revised manuscript. In this study, we found that the wheat–P. huashanica 7Ns disomic addition line 18-1-5 in spike development was significantly faster than its wheat parents from the double ridge stage to the tetrad stage, thereby resulting in earlier heading and maturation in line 18-1-5. We speculate that spike development may play an important role in affecting the heading time of line 18-1-5. Therefore, we performed transcriptome analysis to analyze spike development-specific gene expression patterns for exploring the molecular mechanisms underlying early heading.

Comments 5: The determination of the timing of flowering/heading time is made at the end of the vegetative growth period. The time young spike primordia are formed already entered the reproductive growth phase. Therefore, analysis of gene expression patterns at that time does not reveal the early-heading mechanism. The authors need to analyze gene expression patterns during the transition in leaves from the nutritional growth phase to the reproductive growth phase. Before heading, we observed that the development speed of the young ears of 18-1-5 was all faster than that of the wheat parent line.

Response 5: Many thanks for your insightful comments. We agree with you. The heading stage represents a critical transition period in wheat, marking the shift from vegetative to reproductive growth. Prior to heading, we observed significantly accelerated young spike development in line 18-1-5 compared to its wheat parental lines. Therefore, the expression patterns of spike development-related genes may help elucidate the genetic basis for 18-1-5’s early heading phenotype to a certain extent. Herein, we changed some statements for ‘heading time’ to ‘spike development’ in the revised manuscript.

Reviewer 2 Report

Comments and Suggestions for Authors

Please see comments in the attached word file

Author Response

Dear Reviewer 2:

Thank you very much for taking the time to review our manuscript entitled ‘Transcriptome profiling reveals key genes and pathways associated with early heading in wheat–Psathyrostachys huashanica 7Ns chromosome addition line’ (ID: plants-3681655). Those comments are all valuable and very helpful for revising and improving our paper, as well as the important guiding significance to our researches. We have studied these comments carefully and made the necessary corrections, which we hope will meet with your approval. Revised portion are marked up using the “Track Changes” function in the manuscript. The main corrections in the manuscript and the point-by-point responses to your comments are as follows.

Sincerely yours,

Houyang Kang

Point-by-point response to Reviewer 2

Comments 1: I read with interest the manuscript which I consider deserves publication. However, the text would benefit from some minor changes not only to capture readers’ attention but also to give purpose to it.

Response 1: Thanks very much for your time to review this manuscript. We really appreciate all your comments and suggestions. We have considered these comments carefully and tried our best to address every one of them.

Comments 2: When authors mention in abstract: Developing early-heading wheat cultivars is an important breeding strategy for saving photo-terminal resources, … -what authors want to say with photo-terminal resources? Please make the language clear.

Response 2: Thanks for your insightful comments. We changed this sentence to ‘Developing early-heading wheat cultivars is an important breeding strategy to utilize light and heat resources, facilitate the multiple-cropping systems, and enhance annual grain yield.’ in lines 25-27.

Comments 3: Why Psathyrostachys huashanica Keng (2n = 2x = 14, NsNs) is a potentially useful germplasm of early heading?

Response 3: Many thanks for raising your question. Within the Triticeae tribe, Psathyrostachys huashanica is recognized for its exceptionally early heading date. Its flowering period, occurring from March to April, aligns with that of common wheat but differs from other Triticeae species. Consequently, this species possesses favorable genes associated with early heading and maturation, making it a valuable germplasm resource for breeding early-heading wheat.

Comments 4: Please specify with concrete results the affirmation: Our results increase the understanding of heading time in wheat–P. huashanica 7Ns addition line at the transcriptional level and provide valuable information for further studies on the regulatory mechanism, candidate genes, and genetic resources of early-heading breeding in wheat.

Response 4: As your suggestion, we revised this sentence to ‘Our findings reveal spike development-specific gene expression and critical regulatory pathways associated with early heading in the wheat–P. huashanica 7Ns addition line at the transcriptional level, and provide a new genetic resource for further dissection of the molecular mechanisms underlying heading date in wheat.’ in lines 47-52.

Comments 5: In the affirmation ‘Furthermore, it is also beneficial for promoting wheat-rice or wheat-mize rotations in Southwest China.’ – in my opinion shouldn’t be ‘furthermore’ but instead ‘more importantly’. This is really a very good reason for the study to occur.

Response 5: Thank you for your suggestion. We replaced ‘Furthermore’ by ‘More importantly’ in line 63.

Comments 6: The sentence ‘As an important tertiary gene pool of wheat genetic improvement, Psathyrostachys huashanica Keng ex P. C. Kuo (2n = 2x = 14, NsNs) harbors numerous agronomically beneficial traits, such as early maturity and resistance to biotic and abiotic stresses – should be resumed and added in the abstract (comment 2.)

Response 6: As your suggestion, we added the following sentence in the abstract in lines 28-30: ‘Psathyrostachys huashanica Keng (2n = 2x = 14, NsNs) possesses numerous agronomically beneficial traits for wheat improvement, such as early maturity and resistance to biotic and abiotic stresses.’

Comments 7: The paragraph ‘Interestingly, in the current study, we found that 18-1-5 exhibited earlier heading and maturation than its wheat parents Chinese Spring (CS) and CSph2b through three consecutive years of field investigation. Anatomical observation of young spikes was conducted to further investigate their phenotypic differences. Moreover, comparative transcriptome analysis between 18-1-5 and its wheat parents was performed during four different spike development stages to explore the potential molecular mechanisms.’ - would be better in the abstract

Response 7: Many thanks for your comments. We revised the paragraph to ‘In the current study, we investigated the heading and maturity times of 18-1-5 and its wheat parents, Chinese Spring (CS) and CSph2b, under field conditions. Anatomical observation of young spikes was performed to further characterize their phenotypic differences. Moreover, comparative transcriptome analysis between 18-1-5 and its wheat parents was conducted across four different spike development stages to explore the underlying molecular mechanisms.’ in lines 120-127.

Comments 8: indicate the source of Chinese Spring

Response 8: As your suggestion, we added the source of Chinese Spring in line 550: ‘from Sichuan province, China’.

Comments 9: provide the link for IWGSC RefSeq v1.1

Response 9: As your suggestion, we provided the link for IWGSC RefSeq v1.1: ‘https://wheat-urgi.versailles.inra.fr/Seq-Repository/Annotations’ in line 664.

Comments 10: Please check the availability and provide a more recent date for: TFs using the online tool iTAK (Plant Transcription factor & Protein Kinase Identifier and Classifier, http://itak.feilab.net/cgi-bin/itak/index.cgi, accessed on 25 December 2024) [92].

Response 10: Thank you for your careful work. We carefully checked the availability of the link, and provided a more recent date in line 682: ‘accessed on 5 June 2025’.

Comments 11: Please provide a specific statement for this sentence: Our results provide useful information for future investigations on the molecular mechanisms of wheat heading time regulation. – give example of what kind of investigations can be further done.

Response 11: As your suggestion, we revised this sentence to ‘Our results provide useful information for future investigations into the molecular mechanisms of wheat heading, such as gene function validation and protein-protein interactions.’ in lines 610-711.

Reviewer 3 Report

Comments and Suggestions for Authors

The manuscript is written on a current and interesting topic. It very appropriately combines cytogenetic approaches with transcriptomics. The authors obtained a large amount of information, as evidenced by, among other things, extensive supplementary materials (12 TableS). However, the authors did not always succeed in appropriately implementing the information into current knowledge. I specify my comments and remarks in my review:  

Formally, I can say that the manuscript is at a good level, but on line 61 Eps should be italicized, because it is about a gene. At the same time, I see a problem in places with the overuse of references to Tables. As an example, I could cite lines 302-309 or, in extreme cases, in paragraph lines 324-343, where it would be enough to edit the first sentence of the paragraph and then not use the reference to Table 2 repeatedly (it seems distracting).

Now to the individual parts of the text, i.e. firstly the Introduction. Here, on lines 73-76, the interactions between phytohormones and the time of heading/flowering are presented. I miss MeJA in the list of phytohormones (e.g., the current publication on rice DOI: 10.17221/43/2024-CJGPB, where the profile of several phytohormones was determined). In general, it is appropriate to update some references in this section with more recent ones, as the authors are sometimes 10-20 years behind, while knowledge in this area is developing very dynamically.  

Results - are described adequately. Perhaps with only one formal remark, which I specify above, i.e. repeated use of references to one table/figure after another is frequent in the text even though it is clear that the text refers to the given figure/table.  

Discussion - in section 3.3., where the authors discuss the interaction with phytohormones with heading and flowering time, I would repeat my remark about MeJA from the Introduction section. In section 3.4. the authors devote themselves to genes determining heading time in plants, including rice, where I would recommend implementing more up-to-date information than those published more than 20 years ago, e.g. when looking in the Web of Science I found publications DOI: 10.17221/2/2024-CJGPB and 10.17221/66/2024-CJGPB dedicated to this issue (Hd genes), i.e. from 2024! In general, I would also recommend that authors make a critical comparison of their results with other authors, because in some places the authors were satisfied with just listing information from other authors without comparison. This would also be beneficial due to the increase in the novelty of knowledge in the given area.  

Materials and Methods - I have one major comment here on parts 577-588, where the authors describe the methodology without adequate references to the given procedure with only one reference to the production recommendation. Since this is a major part, there should be references to published procedures for further use of the results, or. linking to the authors' activities by other teams dealing with the same or similar issues. The methodology must be written in such a way that it is repeatable without the possibility of obstacles/unknowns.  

The conclusion is adequate.  

Based on the above comments, I recommend the manuscript for publication after major revision and second review.

Author Response

Dear Reviewer 3:

Thank you very much for taking the time to review our manuscript entitled ‘Transcriptome profiling reveals key genes and pathways associated with early heading in wheat–Psathyrostachys huashanica 7Ns chromosome addition line’ (ID: plants-3681655). Those comments are all valuable and very helpful for revising and improving our paper, as well as the important guiding significance to our researches. We have studied these comments carefully and made the necessary corrections, which we hope will meet with your approval. Revised portion are marked up using the “Track Changes” function in the manuscript. The main corrections in the manuscript and the point-by-point responses to your comments are as follows.

Sincerely yours,

Houyang Kang

Point-by-point response to Reviewer 3

Comments 1: The manuscript is written on a current and interesting topic. It very appropriately combines cytogenetic approaches with transcriptomics. The authors obtained a large amount of information, as evidenced by, among other things, extensive supplementary materials (12 TableS). However, the authors did not always succeed in appropriately implementing the information into current knowledge. I specify my comments and remarks in my review.

Response 1: We are very grateful for the time you dedicated to reviewing our manuscript and for your insightful comments and suggestions. We have carefully considered each one and made every effort to address them in the revised manuscript.

Comments 2: Formally, I can say that the manuscript is at a good level, but on line 61 Eps should be italicized, because it is about a gene. At the same time, I see a problem in places with the overuse of references to Tables. As an example, I could cite lines 302-309 or, in extreme cases, in paragraph lines 324-343, where it would be enough to edit the first sentence of the paragraph and then not use the reference to Table 2 repeatedly (it seems distracting).

Response 2: Thank you for your careful work. We revised ‘Eps’ to Italic format in line 68. Additionally, we removed all the duplicate references for Tables 1 and 2 in lines 311-318 and lines 339-353.

Comments 3: Now to the individual parts of the text, i.e. firstly the Introduction. Here, on lines 73-76, the interactions between phytohormones and the time of heading/flowering are presented. I miss MeJA in the list of phytohormones (e.g., the current publication on rice DOI: 10.17221/43/2024-CJGPB, where the profile of several phytohormones was determined). In general, it is appropriate to update some references in this section with more recent ones, as the authors are sometimes 10-20 years behind, while knowledge in this area is developing very dynamically.  

Response 3: As your suggestion, we added ‘methyl jasmonate (MeJA)’ in line 82 and provided corresponding reference: ‘[16] Yan, Z.; Deng, R.; Tang, H.; Zhang, H.; Zhu, S. Molecular basis of differential sensitivity to MeJA in floret opening between indica and japonica rice. Czech J. Genet. Plant Breed2024, 60, 136-148.’ Furthermore, we also revised the numbers of other literatures.

Comments 4: Results - are described adequately. Perhaps with only one formal remark, which I specify above, i.e. repeated use of references to one table/figure after another is frequent in the text even though it is clear that the text refers to the given figure/table.  

Response 4: Thank you for your careful work. we removed all the duplicate references for tables and figures in the revised manuscript.

Comments 5: Discussion - in section 3.3., where the authors discuss the interaction with phytohormones with heading and flowering time, I would repeat my remark about MeJA from the Introduction section. In section 3.4. the authors devote themselves to genes determining heading time in plants, including rice, where I would recommend implementing more up-to-date information than those published more than 20 years ago, e.g. when looking in the Web of Science I found publications DOI: 10.17221/2/2024-CJGPB and 10.17221/66/2024-CJGPB dedicated to this issue (Hd genes), i.e. from 2024! In general, I would also recommend that authors make a critical comparison of their results with other authors, because in some places the authors were satisfied with just listing information from other authors without comparison. This would also be beneficial due to the increase in the novelty of knowledge in the given area.

Response 5: Many thanks for your insightful comments. We added ‘MeJA’ in line 481. Additionally, we added the following references in lines 531-533 and 909-914: ‘[66] Nasution, K.; Satyawan, D.; Yunus, M.; Dewi, A.; Melati, P.; Maryono, M.; Dwimahyani, I.; Enggarini, W.; Sobrizal, S. Detection of genomic loci associated with days to heading in tropical japonica rice through QTL-seq. Czech J. Genet. Plant Breed. 2025, 61, 23-30.’ and ‘[67] Zhang, H.; Wang, L.; Xie, Y.; Hao, L.; Wang, Z.; Yi, C.; Guo, H.; Gan, Y.; Xiang, G.; Yan, Z.; et al. QTL mapping for heading date and plant height using a RIL population in rice in different photoperiod environments. Czech J. Genet. Plant Breed2024, 60, 119-125.’

Comments 6: Materials and Methods - I have one major comment here on parts 577-588, where the authors describe the methodology without adequate references to the given procedure with only one reference to the production recommendation. Since this is a major part, there should be references to published procedures for further use of the results, or. linking to the authors' activities by other teams dealing with the same or similar issues. The methodology must be written in such a way that it is repeatable without the possibility of obstacles/unknowns.

Response 6: Thank you for your careful work. We added the following reference in lines 653 and 973-975: ‘[92] Liu, Z.; Niu, F.; Yuan, S.; Feng, S.; Li, Y.; Lu, F.; Zhang, T.; Bai, J; Zhao, C.; Zhang, L. Comparative transcriptome analysis reveals key insights into fertility conversion in the thermo-sensitive cytoplasmic male sterile wheat. Int. J. Mol. Sci. 2022, 23, 14354.’

Round 2

Reviewer 1 Report

Comments and Suggestions for Authors

In this study, the authors revealed that wheat–P. huashanica 7Ns disomic chromosome addition line (2n = 44 = 42W + â…¡7Ns) showed earlier heading and earlier maturation than its wheat parents. They found that no significant differences appeared among the materials at the apex elongation stage. Surprisingly, the spike development speed is accelerating in the chromosome addition lines compared to the original wheat line. This is a very interesting finding. The experiments in this study were good and the results are clear. The aim of this study should be to elucidate the mechanism of accelerated spike development speed in chromosome addition lines. For this purpose, the authors have performed an RNA-seq analysis.

In the Introduction, the authors need to clearly summarize the prematurity of wheat. Factors that contribute to early maturing of wheat are:

  1. Earlier timing of conversion to shoot apical meristem into spike meristem (transition from vegetative to reproductive growth). This transition of growth phase is usually referred to as flowering. No significant differences appeared among the materials at the apex elongation stage, meaning that no differences in flowering time!
  2. The speed of ear development from the spike primordium to the formation of mature spikes is rapid. Surprisingly, the spike development speed is accelerating in the chromosome addition lines compared to the original wheat line. It is very interesting!
  3. Short period of time between ear formation in the stem and ear emergence (heading time).
  4. Short period of time between heading time and blooming time (pollen is dispersed from anthers).
  5. Short period of time between blooming time and maturation time.

The authors need to explain the components of this EARLY MATURING in the introduction and clarify the purpose of this study.

In the Results, the authors must clearly communicate to the reader the implications of the study's findings. In other words, the DEGs obtained in this experiment are genes related to the speed of spike development. The authors should clarify this point and describe their results.

In the Discussion, the authors summarize the references and their findings on genes related to spike development in wheat, and it is necessary to declare loudly that in this experiment they were able to analyze the genes involved in the speed of spike formation.

Minor points

Figure 3 is missing scale bars.

The characters in Figure 5 and Figure 6 and Figure 7 are too small to discern.

Author Response

Dear Reviewer 1:

Thank you for your continued engagement with our manuscript entitled “Transcriptome profiling of spike development reveals key genes and pathways associated with early heading in wheat–Psathyrostachys huashanica 7Ns chromosome addition line” (ID: plants-3681655). We sincerely appreciate your time and valuable insights, which have significantly strengthened our work. We have carefully addressed all additional comments and suggestions provided in your latest review, which we hope will meet with your approval. The revised manuscript includes necessary modifications marked with "Track Changes" for your convenience. Below, we provide a point-by-point response to your new feedback:

Response to Reviewer 1’s Additional Comments

Comment 1: In this study, the authors revealed that wheat–P. huashanica 7Ns disomic chromosome addition line (2n = 44 = 42W + â…¡7Ns) showed earlier heading and earlier maturation than its wheat parents. They found that no significant differences appeared among the materials at the apex elongation stage. Surprisingly, the spike development speed is accelerating in the chromosome addition lines compared to the original wheat line. This is a very interesting finding. The experiments in this study were good and the results are clear. The aim of this study should be to elucidate the mechanism of accelerated spike development speed in chromosome addition lines. For this purpose, the authors have performed an RNA-seq analysis.

Response 1: We sincerely appreciate your positive assessment of our findings and valuable suggestion. We fully agree that refining the study's aim to explicitly elucidate the mechanism underlying accelerated spike development speed in the chromosome addition lines significantly enhances the manuscript's scientific impact.

Comment 2: In the Introduction, the authors need to clearly summarize the prematurity of wheat. Factors that contribute to early maturing of wheat are: (1) Earlier timing of conversion to shoot apical meristem into spike meristem (transition from vegetative to reproductive growth). This transition of growth phase is usually referred to as flowering. No significant differences appeared among the materials at the apex elongation stage, meaning that no differences in flowering time. (2) The speed of ear development from the spike primordium to the formation of mature spikes is rapid. Surprisingly, the spike development speed is accelerating in the chromosome addition lines compared to the original wheat line. It is very interesting. (3) Short period of time between ear formation in the stem and ear emergence (heading time). (4) Short period of time between heading time and blooming time (pollen is dispersed from anthers). (5) Short period of time between blooming time and maturation time. The authors need to explain the components of this EARLY MATURING in the introduction and clarify the purpose of this study.

Response 2: Many thanks for your insightful suggestion. We explained the components of early maturation in lines 84-91: “Factors contributing to early wheat maturation include: (1) Accelerated floral transition, characterized by premature conversion of the shoot apical meristem into spike meristem; (2) Expedited spike development with a rapid progression from primordium formation to mature spike; (3) Shortened pre-heading phase between spike formation and ear emergence; (4) Reduced heading-anthesis interval; (5) Abbreviated grain-filling period from anthesis to physiological maturity. Crucially, accelerated spike development and early phase transitions represent the core physiological determinants of this process [21, 22].” The purpose of this study was further clarified in line 128. Additionally, we added the following references in lines 798-801:

[21] McMaster, G.S. Phytomers, phyllochrons, phenology and temperate cereal development. J. Agr. Sci. 2005, 143, 137-150.

[22] Kamran, A.; Iqbal, M.; Spaner, D. Flowering time in wheat (Triticum aestivum L.): a key factor for global adaptability. Euphytica 2014, 197, 1-26.

Comment 3: In the Results, the authors must clearly communicate to the reader the implications of the study's findings. In other words, the DEGs obtained in this experiment are genes related to the speed of spike development. The authors should clarify this point and describe their results.

Response 3: Many thanks for your comments. We have made corresponding modifications as your suggestion.

Comment 4: In the Discussion, the authors summarize the references and their findings on genes related to spike development in wheat, and it is necessary to declare loudly that in this experiment they were able to analyze the genes involved in the speed of spike formation.

Response 4: Many thanks for your comments. We have done.

Comment 5: Figure 3 is missing scale bars.

Response 5: Thank you for your careful work. We added the missing scale bars in Figure 3.

Comment 6: The characters in Figure 5 and Figure 6 and Figure 7 are too small to discern.

Response 6: Thank you for your careful work. We adjusted the character size in Figures 5, 6, and 7.

Reviewer 3 Report

Comments and Suggestions for Authors

The authors accepted all comments. If these were incorporated into the manuscript and adequately justified.
Nevertheless, I would like to point out some minor points (of a more formal nature) that would be appropriate to correct before accepting the manuscript for publication:
line 89 - SD - is used for the first time without explanation;
line 125 - in vitro (italics);
Fig. 2 a and c - is the marked variability SD or SE?;
Fig. 3 - the authors use a "scale bar" in the legend, but it is not visible in their own photographs. It would be appropriate to supplement it or make it more visible;
Fig. 7 - is the variability SD or SE?;
line 676 - Abbreviations - not all are listed;
line 677 and 678 - in situ (italics).
These are minor points, but they will certainly contribute to the very good level of this high-quality and interesting study.
Based on the above, I recommend publishing the manuscript after minor revision.

Author Response

Dear Reviewer 3:

Thank you for your continued engagement with our manuscript entitled “Transcriptome profiling of spike development reveals key genes and pathways associated with early heading in wheat–Psathyrostachys huashanica 7Ns chromosome addition line” (ID: plants-3681655). We sincerely appreciate your time and valuable insights, which have significantly strengthened our work. We have carefully addressed all additional comments and suggestions provided in your latest review, which we hope will meet with your approval. The revised manuscript includes necessary modifications marked with "Track Changes" for your convenience. Below, we provide a point-by-point response to your new feedback:

Response to Reviewer 3’s Additional Comments

Comment 1: The authors accepted all comments. If these were incorporated into the manuscript and adequately justified. Nevertheless, I would like to point out some minor points (of a more formal nature) that would be appropriate to correct before accepting the manuscript for publication.

Response 1: We are sincerely grateful for your acknowledgment of our revisions and your thorough and meticulous review. We thank you for identifying these additional refinements and confirm that all remaining editorial suggestions will be diligently addressed in the final manuscript.

Comment 2: line 89 - SD - is used for the first time without explanation.

Response 2: Thank you for your careful work. We have given the full name of "SD" for the first time in line 67.

Comment 3: line 125 - in situ (italics).

Response 3: Thank you for your careful work. We revised “in situ” to italics in line 134.

Comment 4: Fig. 2 a and c - is the marked variability SD or SE?

Response 4: Thank you for your careful work. We added the following detailed explanation in line 164: “Error bars represent standard deviation”

Comment 5: Fig. 3 - the authors use a "scale bar" in the legend, but it is not visible in their own photographs. It would be appropriate to supplement it or make it more visible.

Response 5: Thank you for your careful work. We added the missing scale bars in Figure 3.

Comment 6: Fig. 7 - is the variability SD or SE?

Response 6: As your suggestion, we added the following detailed explanation in line 375: “Error bars represent standard deviation”

Comment 7: line 676 - Abbreviations - not all are listed.

Response 7: Thank you for your suggestion. We added new abbreviations in lines 690-700.

Comment 8: line 677 and 678 - in situ (italics).

Response 8: Thank you for your careful work. We revised “in situ” to italics in lines 692 and 693.
